# Impairments of Spatial Memory and N-methyl-d-aspartate Receptors and Their Postsynaptic Signaling Molecules in the Hippocampus of Developing Rats Induced by As, Pb, and Mn Mixture Exposure

**DOI:** 10.3390/brainsci13121715

**Published:** 2023-12-14

**Authors:** Lalit P. Chandravanshi, Prashant Agrawal, Hany W. Darwish, Surendra Kumar Trigun

**Affiliations:** 1Department of Forensic Science, Sharda University, Greater Noida 201308, India; chandravanshi04@gmail.com (L.P.C.); prashant.agrawal@sharda.ac.in (P.A.); 2Department of Pharmaceutical Chemistry, College of Pharmacy, King Saud University, P.O. Box 2457, Riyadh 11451, Saudi Arabia; 3Department of Zoology, Institute of Science, Banaras Hindu University, Varanasi 221005, India

**Keywords:** developmental neurotoxicity, NMDA receptors, PSD-95, spatial memory

## Abstract

Exposure to metal mixtures is recognized as a real-life scenario, needing novel studies that can assess their complex effects on brain development. There is still a significant public health concern associated with chronic low levels of metal exposure. In contrast to other metals, these three metals (As, Pb, and Mn) are commonly found in various environmental and industrial contexts. In addition to additive or synergistic interactions, concurrent exposure to this metal mixture may also have neurotoxic effects that differ from those caused by exposure to single components. The NMDA receptor and several important signaling proteins are involved in learning, memory, and synaptic plasticity in the hippocampus, including CaMKII, postsynaptic density protein-95 (PSD-95), synaptic Ras GTPase activating protein (SynGAP), a negative regulator of Ras-MAPK activity, and CREB. We hypothesized that alterations in the above molecular players may contribute to metal mixture developmental neurotoxicity. Thus, the aim of this study was to investigate the effect of these metals and their mixture at low doses (As 4 mg, Pb 4 mg, and Mn 10 mg/kg bw/p.o) on NMDA receptors and their postsynaptic signaling proteins during developing periods (GD6 to PD59) of the rat brain. Rats exposed to As, Pb, and Mn individually or at the same doses in a triple-metal mixture (MM) showed impairments in learning and memory functions in comparison to the control group rats. Declined protein expressions of NR2A, PSD-95, p- CaMKII, and pCREB were observed in the metal mix-exposed rats, while the expression of SynGAP was found to be enhanced in the hippocampus as compared to the controls on PD60. Thereby, our data suggest that alterations in the NMDA receptor complex and postsynaptic signaling proteins could explain the cognitive dysfunctions caused by metal-mixture-induced developmental neurotoxicity in rats. These outcomes indicate that incessant metal mixture exposure may have detrimental consequences on brain development.

## 1. Background

Metals are introduced into the environment as mixtures from various anthropogenic sources such as mining, refining, the smelting of ores, or agriculture and natural activities [1,2]. The exposure of human beings to arsenic (As), lead (Pb), and manganese (Mn) through water and foods has raised increasing concerns about health risks [3,4,5]. The interactions of metals in these mixtures can influence their uptake and deposition in organs, which may eventually amplify the overall toxicity in exposed organisms [6,7,8]. In particular, various epidemiological studies have shown that prenatal exposure to heavy metals like As, Cd, Pb, and mercury (Hg), as well as exposure to essential trace elements like Mn and zinc (Zn), are associated with birth defects, adverse pregnancy outcomes, and cognitive and behavioral dysfunctions in later infancy [9,10,11,12,13].

To date, most studies have been focused largely on single-metal exposures [14,15,16]. However, the U.S. Environmental Protection Agency (USEPA) recommends a greater emphasis on understanding the combined toxic effects of the metal mixture [9,17,18,19]. As, Pb, Mn, mercury (Hg), cadmium (Cd), and other metals are also found in mixtures of different combinations in the environment, especially in drinking water [20]. There is an urgent need to understand how mixtures of metals affect health. Of the various metals, As, Pb, and Mn have frequently been reported as a mixture in the environment, industrial settings, and in foods [21,22,23,24], which are likely to produce a neurotoxic response. Thus, chronic exposure to low doses of such metal mixtures poses a greater risk of neurotoxicity [25]. Therefore, these metals can have negative impacts on neurodevelopment during fetal and early postnatal life [26,27,28,29]. 

To date, information regarding the effects of exposure to metal mixtures, in different combinations, in the developing brain remains to be explored. 

Children are more vulnerable than adults to environmental contaminants as they show immature detoxification mechanisms and consume more water and food per unit of body weight than adults [30]. Metal contaminant exposure during fetal and early life makes children more susceptible than adults, as many of them cross the placenta and blood–brain barrier (BBB) [31,32]. Children living near mining, industrial, and agricultural areas of Spain have reportedly shown increased levels of Cd, Pb, and Mn in urine and As in hair [33,34]. These studies further described that a risk of urinary tract birth defects is a common issue in children exposed to municipal solid waste mainly containing metal mixtures of As, Cd, Cr, Mn, Hg, Ni, and Pb. Associations of in utero exposure to Pb and Mn with neurodevelopment abnormalities at 2 years of age, with respect to cognition and language, have also been observed. Further, a link between the additive effects of exposure to Mn and Pb and their increased levels in cord blood samples was also established [35]. Another environmental study demonstrated that co-exposure to environmental Pb and Mn is negatively associated with the intelligence of school-aged children due to the additive or interactive effects of Pb and Mn during the early-life developmental window [36]. A synergetic impact of As and Mn was also found to be associated with child developmental abnormalities, including their reduced perceptual reasoning and working memory and verbal comprehension scores [37]. The people of Bangladesh are more exposed to As, Mn, and Pb in drinking water, and a link between prenatal-to-early-life exposure to this metal mixture and neurodevelopment abnormalities has also been established [27]. Another semi-ecological finding suggested that As, Cd, Mn, and Pb levels in drinking water were associated with birth defect prevalence in North Carolina [9]. 

Exposure to As and Pb, even at low levels, is found to induce cognitive [38,39] and behavioral deficits [40,41] as well as motor dysfunction [42,43] independently. The development of neurological disorders like Parkinson’s disease due to Mn exposure at a high dose has also been reported [44]. Although Mn, As, and Pb are common environmental neurotoxic metals, the relationship between developmental exposure to this metal mixture at low doses and cognitive dysfunction, including learning and memory, as well as the underlying mechanism, are not clear. Currently, developmental neurotoxicity challenged due to the co-exposure of multiple metals needs to be understood through a suitable experimental plan. This should heavily emphasize the mechanism of the synergetic neurotoxic effects of metals and their interactions during the critical period of brain development. 

The metals As, Pb, and Mn may cause neurotoxicity through different molecular mechanisms by interacting with neurotransmitters, receptors and second messenger systems. Metal mixtures are likely to cause site-specific damage to various brain regions including the hippocampus [45]. The hippocampus is a major brain area that controls cognitive functions, including learning and memory [46]. N-methyl-d-aspartate receptors (NMDARs) are glutamate-gated ion channel receptors, which play key roles in cognitive functions, learning and memory, synaptic plasticity, and excitatory synaptic transmission [47]. NMDARs are composed of NR1, NR2 (NR2A-D), and NR3 (NR3A and NR3B) subunits [48,49]. The functional sensitivity of NMDA receptors depends on glutamate release from the pre-synaptic terminal at excitatory synapses and glutamate binding to the receptor at the postsynaptic site [50]. The interaction of glutamate and NMDA receptors could increase Ca^2+^ influx in the postsynaptic neurons and thereby activate various Ca^2+^-dependent enzymes (calmodulin, phospholipases, kinases) as a part of its signaling pathway, ultimately developing either long-term potentiation (LTP) or long-term depression (LTD) in the hippocampus [51,52]. CREB is a cAMP element binding protein which plays a significant role in long-term memory formation, synaptic plasticity, and memory consolidation [53]. Synaptic plasticity is known to involve the activation of CREB, which is activated by different pathways [54]. After activation, CREB translocates into the nucleus and increases the expression of the target genes involved in neurotransmitter release and in restructuring synaptic connections. PSD-95 is another protein involved in NMDAR-mediated signaling and thereby regulates the NMDAR channel opening rate and the number of functional channels, and it is required to sustain the molecular organization of postsynaptic density [55,56,57]. SynGAP is highly expressed at excitatory synapses in the hippocampus coupled with postsynaptic density 95 (PSD-95) and NMDARs [58,59]. SynGAP also negatively regulates Ras signaling, such as the MAP kinase pathway at excitatory synapses, and may regulate the induction of LTP and synaptic plasticity [36]. Calcium/calmodulin-dependent protein kinase II (CaMKII) is highly concentrated in postsynaptic density (PSD) and gets activated by Ca^2+^ influx through NMDARs during the induction of LTP [60]. Activated CaMKII can phosphorylate SynGAP and alpha-amino-3-hydroxy-5-methyl-4-isoxazolepropionic acid receptors (AMPARs) during the increased synaptic efficacy of LTP [61]. 

The timing of exposure of the nervous system is very critical as the neurotoxic susceptibility of the brain depends on the stage of brain development. The maternal and early-life periods in rats are critical and crucial for brain development as structural and functional maturation takes place primarily during this time. The period from GD6 (gestational day) to PD60 (post-lactational day) is the developmental period in rats during which the maturation of different neurotransmitter systems, receptors, the BBB, the setting of neurons, and neurochemical synaptogenesis takes place [62]. However, the impact of As, Pb, and Mn mixture exposure on learning memory and NMDAR-associated postsynaptic signaling players during this period of brain development remains largely unexplored.

Therefore, the present study was carried out to determine the synergetic effects of As, Pb, and Mn mixture exposure during the critical period of brain development by evaluating spatial learning and memory functions vs. profiles of the NMDAR subunits and their signaling players at the PD60 stage.

## 2. Materials and Methods

### 2.1. Animals and Treatment

The Animal House facility of CSIR-Indian Institute of Toxicology Research, Lucknow, provided Wistar rats of proven fertility (the males were 12 weeks old and the females were 10 weeks old). A commercial pellet diet was fed to the animals with free access to water. They were housed in plastic cages with bedding in a temperature-controlled room (22 °C/20 °F) with a 12/12 h light/dark cycle. The Institutional Animal Ethics Committee of Banaras Hindu University, Varanasi, India (F.Sc/IAEC/2016-17/233), approved this study as per the international care guidelines (Ministry of Environment and Forests, Government of India). One adult female rat (Wistar strain) was placed with one male in each cage for mating purposes. The appearance of a vaginal plug was investigated for mating and designated as gestation day 0 (GD0). Further, pregnant rats were separated from the cages and were divided into five groups (6 pregnant rats in each group). During the gestation to lactation period, all pregnant rats were housed individually.

Rats were treated from the gestational–lactational period to early postnatal life (GD6 to PD59) as follows: Normal saline was administered to the control rats (group I); rats in groups II, III, and IV were administrated sodium arsenite (Sigma-Aldrich, Saint Louis, MO, USA) (4 mg/kg, p.o. body weight), lead acetate (Sigma-Aldrich, Saint Louis, MO, USA) (4 mg/kg, p.o. body weight), and manganese chloride (Sisco Research Laboratories Pvt. Ltd., Mumbai, India) (10 mg/kg, p.o. body weight) in drinking water, respectively; meanwhile, group V received a metal mixture (As, Mn, and Pb) with the same doses of group II, III, and IV rats. All pregnant rats naturally delivered their pups, which were reared up to weaning until PD 21. The litters were culled to 8 pups/litter in all cases on parturition. The male pups were exposed to metals and their mixture during the gestational–lactational period (GD6 to PD21) through the mother. On PD22, male pups were separated from each group and exposed to the same treatment as their mothers from weaning to adulthood (PD22 to PD59). The effects on behavioral parameters for learning and memory were studied 24 h after the last dose of treatment on PD60.

The control group and metal mixture-exposed rats were sacrificed on PD60, and their brains were dissected from the skull rapidly and washed in ice-cold saline. The brain’s hippocampal region was dissected using the method defined by Glowinski and Iversen, 1966 [63]. The brain regions were kept frozen at −80 °C for neurochemical studies. 

### 2.2. Neurobehavioral Studies

#### 2.2.1. Spatial Learning and Memory Performance by Morris Water Maze test

To evaluate the effect of the metals and their mixtures on cognitive functions, a spatial memory test was performed on the rats as previously described [64]. In this process, the acquisition and spatial localization of important visual clues are processed, consolidated, and eventually recovered in order to successfully navigate and locate a hidden platform to escape water. This was performed in a circular tank of water (170 cm wide, 60 cm high), wherein the floor of a tank was separated into four quadrants designated as Q1, Q2, Q3, and Q4. A detachable platform measuring 11 cm in diameter was positioned in the middle of quadrant Q4, which was submerged at a depth of 2 cm below the water’s surface (23 °C), to serve as a reference memory tool. On the wall of every quadrant, visual cues were placed. For the hidden-platform acquisition, the rats were trained for four trials per day for 5 consecutive days. After being dropped into the water tank, the rat was left to swim until it found the platform within 120 s. The rats were allowed to rest on the platform for 30 s before the next trial. After the hidden-platform acquisition test, the platform was removed, and the rats were allowed to freely swim in the water for 120 s. As a retrieval index, the average spent time finding Q4, the missing platform, was noted. The mean time spent in Q4 in search of the missing platform was recorded as an index of retrieval. An overhead camera was placed to record the testing sessions for analysis with ANY-maze software (Microsoft version 4.84, Stoelting Co., Wood Dale, IL, USA).

#### 2.2.2. Continuous Alternation and Spatial Memory in Y-Maze 

The rats’ spatial working memory was evaluated with the continuous alternation task by using a Y-maze, as previously described [65]. The Y-maze was made of wood and consisted of three arms (40 cm long × 15 cm wide × 30 cm high) with an angle of 120° between each of the two arms and an equilateral triangular central area with 15 cm at its longest axis. The rats were placed at the end of one arm and allowed to move freely through the maze for 5 min without the use of any reinforcer, motivation, or punishment. The ceiling-mounted CCD camera captured the series and order of entry into each arm of the Y-maze. Additionally, the order of arm movements was calculated manually, and the percentage of alternation was determined by dividing the number of alternations (entry into three distinct arms consecutively) by the total number of alternations (the number of arms entered minus 2). The Y-maze test was performed after the exposure on PD60 to assess the spatial memory through novelty-seeking behavior, as previously described [66]. During the first trial (training), the novel arm was blocked and one of the other two arms was randomly designated as a start. The rats were placed at the end of the start arm and allowed to explore the start arm and other arm for 15 min. After an inter-trial interval of 4 h, all three arms were made accessible during the second trial. In the second trial, rats were placed in the same arm (start arm) and given 5 min to explore all arms. Every trial was captured on video using a CCD camera that was positioned in the ceiling. ANY-maze software (Microsoft version 4.84, USA) was used to analyze the video recordings. The results are shown as a percentage of time and entries into the novel arm relative to the other arm. 

#### 2.2.3. Memory in a Passive Avoidance System

This is a fear-motivated system classically used to evaluate the memory of rats. The passive avoidance system consisted of two partition shuttle boxes with a light and a darker one containing a shock generator device, and the test procedure we followed was the same as previously described [67]. During the acquisition phase (first trial), the animal was placed in the lit chamber of the shuttle box. When the rat naturally crossed into the dark chamber, the guillotine door was closed, and the rat received a low-intensity foot shock (0.5 mA, 10 s). The animal learns that going into the dark partition has negative consequences during the acquisition phase.

During the retention trial, the rat was placed in the white chamber after 24 h of acquisition, and the latency time between placement and entry into the dark chamber was measured (maximum 300 s). The shock was not given to the rats in the retention trials to avoid reacquisition. A decrease in the transfer latency time on the retention trial (second trial and higher) as compared to the acquisition trial was considered an impairment in memory.

#### 2.2.4. Learning in an Active Avoidance System

A two-way active avoidance task was performed in a shuttle-box to assess the learning activity of rats following the standard protocol as previously described [38]. In the active avoidance task, animals were placed in a two-partition shuttle box and were required to learn the association between a conditioned stimulus (buzzer) and an unconditioned stimulus (foot shock). Rats were placed individually in one chamber of the shuttle box for habituation for 5 min and allowed to freely explore inside the shuttle box. A warning signal in the form of a buzzer for 10 s was followed by a buzzer and foot shock (0.5 mA for 10 s) to each rat. When the rats had to change the chamber to avoid an electric foot shock during the conditioned stimulus presentation, it was considered a positive conditioned avoidance response. There were 20 trials conducted daily in all exposed and control group rats for 3 days. The number of responses of each animal was recorded. The inter-trial interval was 1 min. Of 20 trials per day, the total number of avoidances and the percentage of conditioned avoidance responses were calculated as a measure of learning ability.

### 2.3. Neurochemical Studies

#### 2.3.1. mRNA Expression of Hippocampal NR2A, NR2B, and NR1 Genes

The gene expression of NR1, NR2A, and NR2B in the hippocampal tissue of rats was measured by RT-PCR (Applied Biosystems, Thermo Scientific, Waltham, MA, USA) with glyceraldehyde 3-phosphated dehydrogenase (GAPDH) used as a normalization marker. cDNA was amplified by using RT-PCR (ABI SYBER green qPCR Kit, Thermo Scientific, Waltham, MA USA), with the help of a 7900HT Fast Real-Time PCR System run at 40 cycles following the supplier protocol: first cycle of denaturation (95 °C, 10 s), followed by 40 cycles of denaturation (95 °C, 10 s), annealing (60 °C, 10 s), and extension (72 °C, 45 s). A final extension (72 °C, 5 min) was then performed. The primer sequences used for NR1, NR2A [68], NR2B [69], and GAPDH [70] are mentioned in Table 1.

#### 2.3.2. Immunodetection of Proteins by Western Blotting

To eliminate the insoluble material, hippocampal tissue was homogenized in RIPA buffer and centrifuged at 12,000× *g* for 15 min at 4 °C following a reported protocol [38]. The supernatant was collected and samples were separated on 10% SDS-PAGE with 30 μg of protein per lane. The proteins were transferred on the nitrocellulose membranes by using electro-blotter and treated with blocking buffer (5% BSA). The proteins on the membranes were incubated overnight at 4 °C with the antibodies NR1, NR2A, NR2B, pCAMKII, pCREB, PSD95, and SynGAP (Cell Signaling Technology, Danvers, MA, USA) followed by incubating them with HRP-conjugated secondary antibodies (Genei, Bangalore, India) (1:4000). The detection of proteins was performed by using an ECL Western blotting detection kit (Pierce, Thermo Scientific, Waltham, MA, USA). The semi-quantification of immunoreactivity was performed with a digital gel image analysis system (Image Quant LAS 500). β-actin was used as a normalization marker.

#### 2.3.3. Estimation of As, Pb, and Mn Levels in the Hippocampus

Rat hippocampal levels of As, Pb, and Mn were quantified using inductively coupled plasma optical emission spectrometry (ICP-OES, Perkin Elmer, Shelton, CT, USA) following a previously published method [71] with minor modifications. Briefly, tissue (approximately 100 mg) was placed in a Kjeldahl flask, and concentrated nitric acid (1 mL) and perchloric acid (1 mL) were added to it. The sample was digested continuously in a sand bath until the solution turned yellow. If the brown color of the digested sample persisted, we added more nitric acid into it for further oxidation. ICP-OES was used to quantify As, Pb, and Mn using aliquots after the digestion was brought up to a suitable volume with deionized water. A calibration curve was drawn by adding known concentrations of As, Pb, and Mn in the range of 0.005 mg/L to 1.0 mg/L in 0.2% (*v*/*v*) nitric acid. The metals were quantified by ICP-OES and data were presented in ng/gm tissue weight.

#### 2.3.4. Protein Estimation

Utilizing bovine serum albumin as a reference standard, the protein content of the samples was determined by following the method previously reported [72].

#### 2.3.5. Statistical Analysis

Using the Graph Pad Prism program, the data were analyzed using one-way analysis of variance (ANOVA) and the Newman–Keuls test for post hoc comparisons. Significant values were defined as *p* < 0.05.

## 3. Results

### 3.1. Neurobehavioral Studies

#### 3.1.1. The Effect on Spatial Memory and Learning: Morris Water Maze Analysis

In the test of MWM for working memory, all rats of the experimental and control groups were trained to find the location of the hidden platform over 5 days of training. Day-by-day decreases in the platform acquisition times across all groups indicated that the rats had acquired space allocation memory, as seen by the reduced latency to locate the platform underwater with repeated training. After the exposure, the rats in the As, Pb, and MM groups exhibited a significant deficit (*p* < 0.05) in hidden-platform acquisition compared to the control groups (Figure 1b). However, the Mn rats exhibited an insignificant deficit (*p* > 0.05) in hidden-platform acquisition. There were no significant differences observed in the spatial probe test among all groups of rats after the metal exposures.

#### 3.1.2. Effect on Continuous Alternation and Spatial Memory in the Y-Maze

Furthermore, the effect on working memory was assessed using a Y-maze after the exposure to metals. The metal-unexposed rats performed the continuous alternation task with 71% accuracy. In contrast, As, Pb, Mn, and MM group rats showed significantly lower mean scores (44%, 54%, 58%, and 33%, respectively) than their control counterparts (Figure 2A). Also, working memory was found to have declined more in the metal mixture-treated rats than the other group; however, no significant change was observed in the Mn-alone group. 

We also analyzed their preference in the start versus novel arms in a Y-maze system. We observed that the percentage of entries (Figure 2B) and time spent (Figure 2C) for the unexposed and Mn-treated rats visiting the novel arm were significantly higher than for the start arms (*p* < 0.05). However, the percentage of entries and time spent for As, Pb, and MM-treated rats exploring the novel arm were not significantly different from the scores obtained for other arms (*p* > 0.05). This shows that the spatial recognition memory of the MM-exposed rats was more impaired than the other treated groups. 

#### 3.1.3. Effect on Memory in the Passive Avoidance System

All treated groups of rats were found to move into the dark chambers early, except the Mn group, indicating an impairment in learning and memory functions. The control (*p* < 0.001) and Mn (*p* < 0.01) groups showed significantly increased transfer latency time (TLT) in the retention trials after 72 h of acquisition trials. On the other hand, a decreased TLT in the retention trials was observed in case of the As, Pb, and MM-treated rats as compared to the acquisition trials. TLT in the retention trial was more compromised in MM (*p* < 0.001), As (*p* < 0.001) and Pb (*p* < 0.001)-exposed rats as compared to the controls (Figure 3A). Moreover, the transfer latency time of the rats was insignificant in the acquisition trial in all the groups.

#### 3.1.4. Effect on Learning Ability in the Active Avoidance System

An active avoidance test, another parameter to assess the learning and memory of rats, was performed, and the results are presented in Figure 3B. An impairment in learning and memory was observed in rats exposed to As (*p* < 0.01), Pb (*p* < 0.05), Mn (*p* > 0.05), and MM (*p* < 0.001) from GD6 to PD59 on PD60 as compared to the controls (Figure 3B). The decrease in learning and memory was more pronounced in the rats treated with the metal mixture.

### 3.2. Neurochemical Studies

#### 3.2.1. Effect on the mRNA Expression of NR2A, NR2B, and NR1 Genes

A significant decrease in the mRNA levels of the NR2A subunit of the NMDA receptor was observed in the hippocampus of all the exposed-group rats, except the Mn group, after the metal exposure as compared to their control counterparts. The reduced mRNA level of NR2A was more evident than the other NMDAR subunits. There were no significant changes observed in the mRNA levels of NR2B and NR1 in the individual metal and metal mixture-exposed rats on PD60 as compared to the control group rats (Figure 4A).

#### 3.2.2. Effect on the Protein Expression of NNMDARs Subunits (NR2A, NR2B, and NR1) and Their Postsynaptic Signaling Proteins (PSD-95, SynGAP, pCAMKII, and pCREB)

NR2A and NR2B subunits of NMDAR can affect long-term potentiation (LTP) and long-term depression (LTD) differently. The present results present changes in NMDAR subunit expression during the critical period of brain development induced by exposure to As, Pb, Mn, and their mixture.

NR2A protein expression was found to be significantly reduced in the hippocampus of As (*p* < 0.001), Pb (*p* < 0.001), and MM (*p* < 0.001)-exposed rats as compared to the control group on PD60 (Figure 4B). However, no significant effect was observed in the Mn (*p* > 0.05) group rats as compared to the control rats. These changes may represent a spatial working memory deficit in the metal-exposed rats. There was no noticeable change observed in the expression of NR2B and NR1 proteins in all groups. However, the expression of NR1 was found to decrease in the MM (*p* < 0.01)-exposed rats due to the synergetic effect of metals as compared to their control counterparts on PD60 (Figure 4B). 

The expression of PSD-95 was significantly reduced in the hippocampus of rats in the As (*p* > 0.05) and MM (*p* < 0.001) groups, respectively. Moreover, there was no significant change observed in the Mn and Pb groups as compared to the control group rats (Figure 5A). Upregulation of SynGAP was observed in all the metal-treated groups, except the Mn group, as compared to the controls. Interestingly, the upregulation of SynGAP was more pronounced in the As (*p* < 0.001) and Pb (*p* < 0.001)-treated groups than the MM group (*p* < 0.01) (Figure 5B). Modifications in the expression of PSD 95 and SynGAP proteins may impair the synaptic plasticity and consolidation or reconsolidation of memory of rats, respectively. 

The functional plasticity of LTP and the processes of learning and memory formation are associated with the activity of pCaMKIIα and pCREB proteins. The expression level of pCaMKIIα was found to be reduced in the hippocampus of As (*p* < 0.05) and MM (*p* < 0.001)-exposed rats as compared to the control group on PD60 (Figure 6A). On the other hand, the expression of pCaMKIIα was not affected by Pb and Mn exposure. As compared to the control rats, As (*p* < 0.01), Pb (*p* < 0.05), Mn (*p* < 0.01), and MM (*p* < 0.001) exposure significantly reduced the expression of pCREB in the hippocampus of rats on PD60 (Figure 6B).

#### 3.2.3. As, Pb, and Mn Levels in the Hippocampus 

As, Pb, and Mn concentrations were estimated in the hippocampus of rats after the metal exposure. Significant (*p* < 0.01 and *p* < 0.001) increases in the levels of all metals were observed in the hippocampus of individual metal-treated rats as well as in the metal-mixture-exposed rats on PD60 as compared to the control rats (Figure 7A–C). Interestingly. the levels of Pb were significantly higher in the MM group as compared to the Pb-alone group rats (*p* < 0.05) (Figure 7B).

## 4. Discussion

Metal mixture exposure may be a decisive cause of cognitive dysfunctions and other neurological disorders in children [73,74,75]. In real-life situations, generally, humans are exposed to metal mixtures instead of single metals [76]. The present study aimed to investigate the effect of long-term low-level metal mixture exposure on NMDARs and its postsynaptic signaling mechanism in neurobehavioral outcomes in rats. This study is close to the scenarios of real-life human exposure and, therefore, it is of great relevance. 

Metal mixture exposure takes place in different stages of life (perinatal, early life, and the later stage of life), amongst which perinatal and early-life exposure is considered to be of more concern. Neurotoxic metals such as As, Pb, and Mn can cross the placental barrier and the BBB [77,78,79,80], thereby placing the developing brain at risk. Altered integrity of the BBB is one of the basic mechanisms via which metals may cause neurological problems [81,82]. We found that As, Pb, and Mn levels in the hippocampus of exposed rats were significantly higher than those in the control group rats. In the metal mixture group, Pb concentrations were significantly higher than the Pb alone-treated animals, suggesting that the presence of As and Mn is likely to alter the deposition of Pb in the hippocampus of rats [83]. Moreover, the synergistic or additive neurotoxic effects of these metals should be considered in “real life” scenarios with regard to brain development. The accumulation of these metals in the hippocampus of rats thus necessitates studying the neurotoxic effect of these metals on neurobehavioral impairments and the associated neurochemical aberrations, focusing on NMDAR postsynaptic signaling.

In the present study, learning and spatial memory were found to be more impaired in MM-exposed rats as opposed to those exposed to single metals. In some studies, it has been argued that neurobehavioral dysfunctions may be linked to alterations in NMDAR postsynaptic signaling in the hippocampus [84]. Impairment in spatial memory and learning was measured by using MWM, Y-maze, and shuttle box tasks. In this study, the performances of perinatally and early life-exposed rats to As, Pb, and MM during the acquisition phase in the MWM were significantly different from the control group rats, thereby suggesting that spatial memory and learning ability were altered on PD60 due to these metal exposures. Working memory and temporary information are stored in the hippocampus and prefrontal cortex [85,86] of the brain. Furthermore, in this study, continuous alternation, a well-known Y-maze test to assess working memory in rats, suggests that working memory was significantly altered in As and MM-exposed rats on PD60. During the two-trial Y-maze task, our data demonstrated that As, Pb, and MM impaired the spatial recognition memory of rats, but the impairment in the spatial memory was more prominent in MM-exposed rats. The animals exposed to these metals and their mixture also showed poor learning and impaired memory during the active and passive avoidance tasks. Impairment in learning memory clearly suggested that, even at low doses, these metal mixtures are able to affect the hippocampus adversely. This decline in the memory ability of rats following exposure to As, Pb, Mn, and MM might be related to the accumulation of these metals in the hippocampus [83]. So far, there are few studies concerning the adverse effects of individual metals, even at low doses, on the learning and memory of rats during perinatal and early-life exposure [38,87,88,89]. This is the first study on rats which explored metal mixture-induced developmental neurotoxicity at low doses on PD60.

Metal mixtures majorly cause more site-specific damage to the hippocampus than the other brain regions [90]. In normal physiological conditions, NMDARs and associated postsynaptic molecules play important roles in learning and memory functions by modulating neuronal synaptic plasticity [91]. The findings herein—of decreased NR2A expression in the hippocampus of As, Pb, and Mn-mixture exposed rats on PD60, but little effect on the other subunits as compared to their control counterparts—suggest that the NR2A subunit is more sensitive to metal-induced developmental neurotoxicity. The pharmacological and signaling properties of the NMDARs are influenced by the subunits in pathological conditions [92]. NR2A is the predominant subunit that plays central roles in LTP in the hippocampus [93]. Additionally, the expression level of NR2A is associated with the level of glutamate (Glu) in the synaptic cleft [94] and oxidative stress [95] in the developing brain and, thus, it becomes further disturbed due to metal exposure. A modification of NMDARs would influence postsynaptic protein activation, wherein the influx of Ca^+2^ at the postsynaptic site becomes the main initiator of further neurochemical changes due to the interaction of Glu-NMDARs by activating the multiple Ca^+2^-dependent enzymes (kinases, calmodulin, phospholipases), which are essential for memory and synaptic plasticity [96].

CREB signaling plays imperative roles in learning, memory, and synaptic plasticity [97,98]. Alterations in the phosphorylation level of CREBs during developmental metal neurotoxicity are yet to be clarified. During the insult period, declined expression of pCREB in all metal-exposed groups suggests a link between the level of pCREB and injury to the hippocampal neurons. NR2A-dependent CREB activation has already been reported [99,100]. The activation of CaMKII, a calcium-dependent protein that co-localizes at the postsynaptic site with NMDRs, suggests it serves as the memory-related signaling molecule for synaptic plasticity [101,102]. The declined expression of pCaMKII in the present study indicates that the neuronal damage is more prominent in As and MM-exposed rats, even at low metal doses, than the other groups. Such effects of low metal doses on the hippocampus of the developing brain and the interaction between pCaMKII and the NMDAR subunit NR2A have been shown to be critically important for affirming the roles of NMDAR postsynaptic signaling in the mechanism of MM-induced developmental neurotoxicity [84,103]. The metal dose-dependent association of SynGAP and PSD95 alterations with NMDAR changes in synaptic plasticity and spatial memory has already been described [104]. In line with this, to explore the effect of MMs on PSD95 in the hippocampus of rats, we observed a declined expression of PSD95, thereby suggesting that MMs could impair the NMDAR-related multiple downstream signaling associated with learning and memory functions. The close association between SynGAP and NMDARs further suggests that SynGAP may have a significant role in the NMDAR-dependent activation of the MAPK/ERK pathway. Further, it is argued that SynGAP negatively regulates the ERK pathway and exhibits a different role in LTP, synaptic plasticity, and learning mechanisms [104,105]. An elevation of SynGAP level was observed in the metal-exposed rats, suggesting that SynGAP inhibited the activity of MAPK/ERK by decreasing the p-ERK1/2 protein level in the hippocampus of the metal mixture-exposed rats. The data of pERK expression are mentioned in the Appendix A. Additionally, the activation of CaMKII inhibits SynGAP [106]. This study demonstrated that increased SynGAP expression and decreased CaMKII activity could inhibit MAPK/ERK activation, as seen from the lower p-ERK1/2 level in the hippocampal regions of rats exposed to MMs. Other studies have also reported a reduced expression of pERK in the hippocampus of As-exposed rats showing cognitive dysfunctions [78,84]. Our findings thus establish an association between metal mixture-induced impairments in NMDAR composition and the postsynaptic signaling molecules (pCREB, pCaMKII, PSD-95, SynGAP) vis-à-vis cognitive dysfunctions, including learning and memory functions, in rats. 

In conclusion, studying the effect of continuous low-level exposure to many combined metals on brain development, during the long developmental period of mammals, is of high scientific concern, as compared to single-metal studies. The present study was an attempt to explore the effect of the most common metal mix (As, pb, and Mn), found as a water pollutant in local vicinities, on the developing brain in a rat model. The findings suggest that in comparison to the effects of these individual metals, low-level MM exposure could cause more intense impairments in spatial memory and learning functions in newborn MM-exposed pups. Such neurobehavioral impairments could also be associated with significant alterations in the level of NMDAR subunits and the postsynaptic density protein PSD95 vis-à-vis changes in the main signaling components of this pathway in the hippocampus, involved in memory consolidation and synaptic plasticity functions, of rats exposed to MMs during the critical period of brain development. The findings thus provide evidence that low-dose but persistent exposure to metal mixtures has a greater impact on developmental neurotoxicity, and hence, this aspect needs to be re-addressed for discerning metal contamination-dependent developing-brain defects in mammals. 

## Figures and Tables

**Figure 1 brainsci-13-01715-f001:**
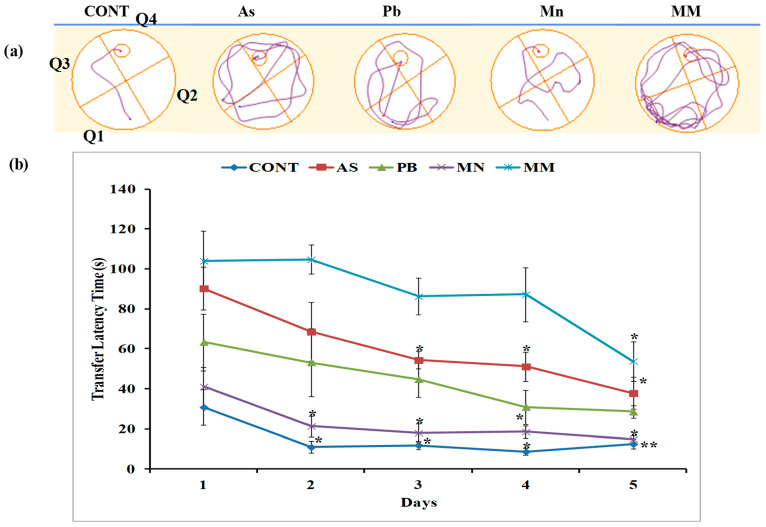
(**a**) The swimming tracks of hidden-platform acquisition on fifth day. The diagram of the tracks shows that the control and Mn group rats easily found the platform, whereas the rats from the As, Pb, and MM groups could find the platform through repeated exploration. Graph (**b**) presents the transfer latency time during the trial from day one to five. As, Pb, and MM groups exhibit significant deficit in hidden-platform acquisition as compared to the controls on PD60. Values are mean ± SEM of five animals in each group. Data were analyzed by one-way analysis of variance followed by Newman–Keuls test. Significantly different: * *p* < 0.05, ** *p* < 0.01 (day 1 vs. rest of the days).

**Figure 2 brainsci-13-01715-f002:**
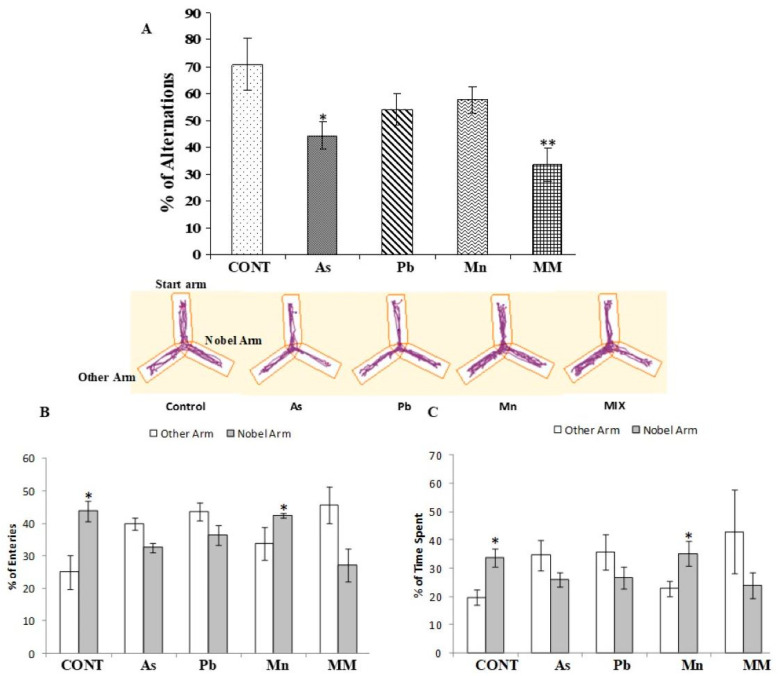
The Y-maze was used to evaluate the impact of exposure to As, Pb, Mn, and their mixture on spatial learning and memory of PD60 rats. During continuous alternation test, (**A**) represents percentage of entries into other/novel arms; (**B**,**C**) show percentage of time spent in other/novel arms. Values are the mean ± SEM of five animals in each group. Significantly different: * *p* < 0.05, ** *p* < 0.01 compared to the control group.

**Figure 3 brainsci-13-01715-f003:**
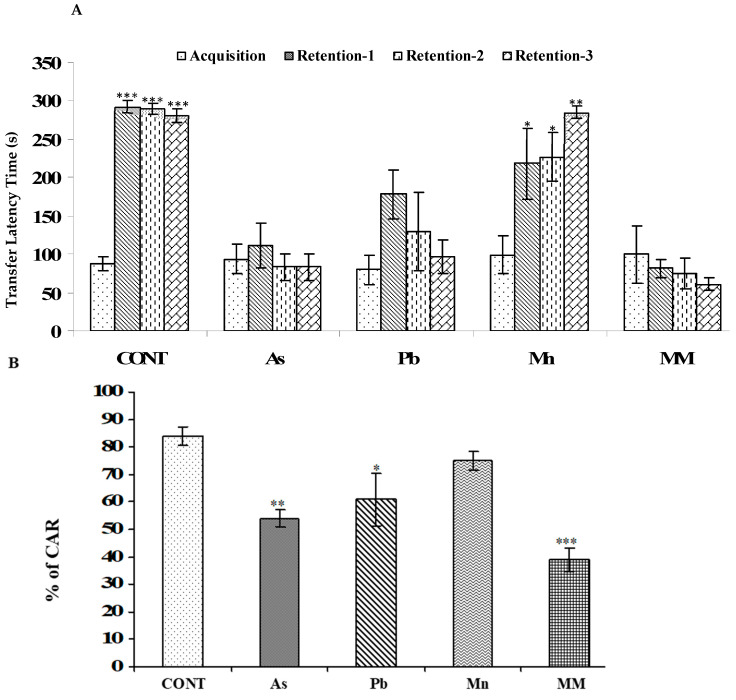
Effect on learning and memory was assessed by passive avoidance (**A**) and active avoidance response systems (**B**) in rats following exposure (GD6-PD59) to As, Pb, Mn, and their mixture on PD60. The passive avoidance response in the retention tests was assessed at various times (24, 48, and 72 h, respectively) after the acquisition trial for rats exposed to As, Pb, Mn, and their mixture. (**A**,**B**) represents the % of condition avoidance response. Values are mean ± SEM of five animals in each group of both tests. Data were analyzed by one-way analysis of variance followed by Newman–Keuls test. Significantly different: * *p* < 0.05, ** *p* < 0.01, *** *p* < 0.001, as compared to controls (**A**). Significantly different: * *p* < 0.05, ** *p* < 0.01, *** *p* < 0.001, as compared to acquisition trial (**B**).

**Figure 4 brainsci-13-01715-f004:**
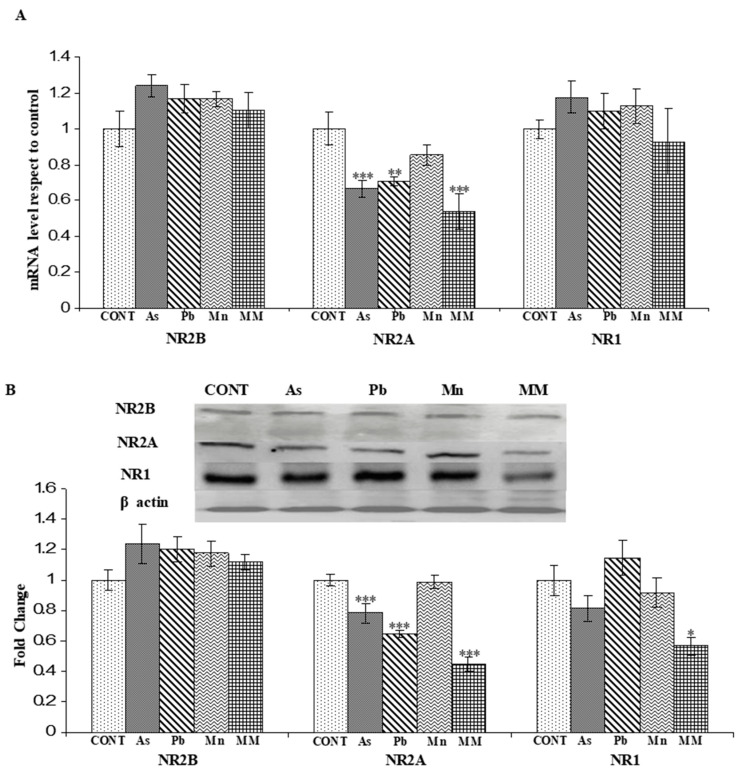
(**A**) The mRNA levels of NR1, NR2A, and NR2B in the hippocampus after the exposure to metals on PD60. Protein expression of NR1, NR2A, and NR2B subunits of NMDAR, in the hippocampus of rats after exposure to As, Pb, Mn, and their mixture (GD6-PD59), was analyzed on PD60 (**B**). The expression levels of NR1, NR2A, and NR2B were normalized against β-actin. Values are mean ± SEM of five animals in each group in the data presented for mRNA levels (**A**) and three animals in each group in the data presented for protein expression (**B**). Data were analyzed by one-way analysis of variance followed by Newman–Keuls test. Significantly different: * *p* < 0.05, ** *p* < 0.01, *** *p* < 0.001 as compared to controls.

**Figure 5 brainsci-13-01715-f005:**
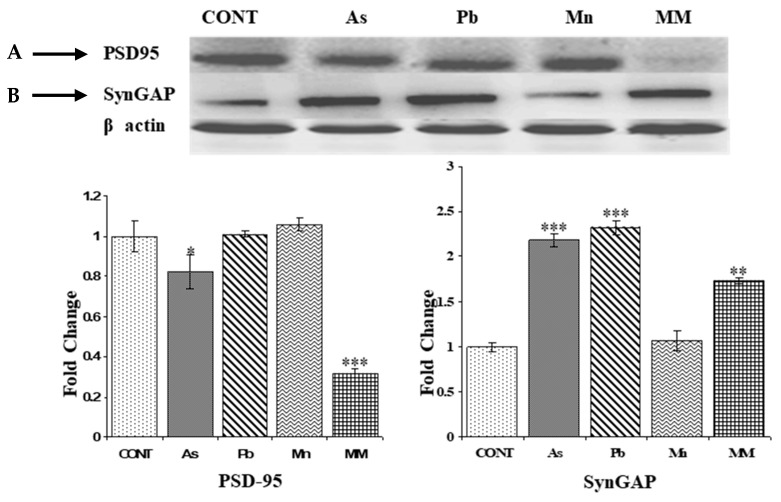
Protein expression of PSD-95 (**A**) and SynGAP (**B**) in the hippocampus of rats after exposure to As, Pb, Mn, and their mixture (GD6-PD59) analyzed on PD60. The expression levels of PSD-95 and SynGAP were normalized against β-actin. Values are the mean ± SEM of three animals in each group. Data have been analyzed by one-way analysis of variance followed by Newman–Keuls test. Significantly different * *p* < 0.05, ** *p* < 0.01, *** *p* < 0.001 as compared to controls.

**Figure 6 brainsci-13-01715-f006:**
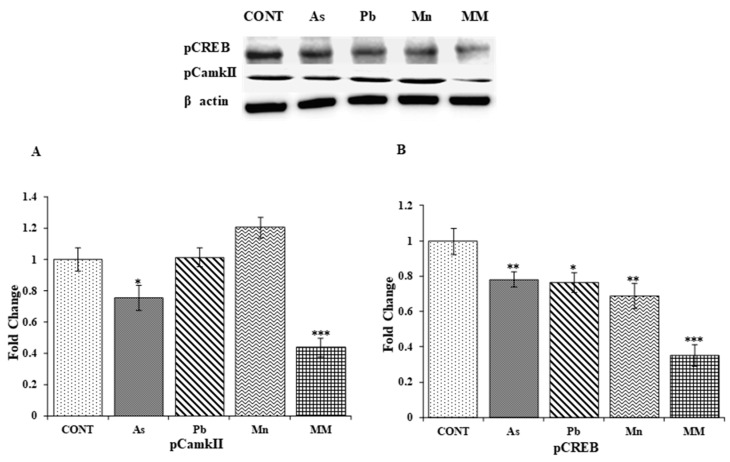
Protein expression of pCamkII (**A**) and pCREB (**B**) in the hippocampus of rats after exposure to As, Pb, Mn, and their mixture (GD6-PD59) analyzed on PD60. The expression levels of pCamkII and pCREB were normalized against β-actin. Values are the mean ± SEM of three animals in each group. Data were analyzed by one-way analysis of variance followed by Newman–Keuls test. Significantly different: * *p* < 0.05, ** *p* < 0.01, *** *p* < 0.001 as compared to controls.

**Figure 7 brainsci-13-01715-f007:**
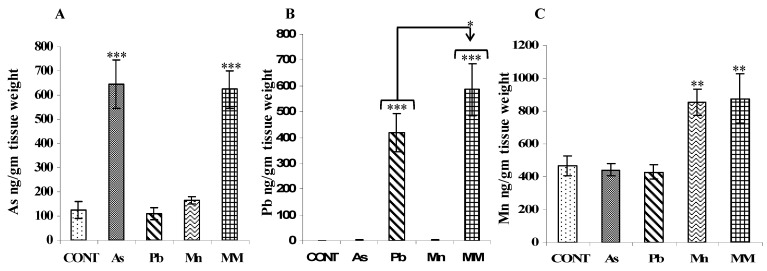
As (**A**), Pb (**B**), and Mn (**C**) levels in the hippocampus were measured immediately after exposure on PD60 by ICP-OES. Metal levels are expressed as ng/gm tissue weight. Values are mean ± SEM of four to five animals in each group. Data were analyzed by one-way analysis of variance followed by Newman–Keuls test. Significantly different: * *p* < 0.05, ** *p* < 0.01, *** *p* < 0.001 as compared to controls.

**Table 1 brainsci-13-01715-t001:** Details of primer sequences of NR1, NR2A, NR2B, and GAPDH.

NR1	FP:5′-TCCACCAAGAGCCCTTCGTG-3′	RP; 5′-AGTTCAACAATCCGAAAAGCTGA-3′
NR2A	FP:5′-GACTGGGACTACAGCCTG-3′	RP; 5′-CTTCTCTGCCTGCCCATAGC-3′
NR2B	FP:5′-GCCGGCAGCATTCCTACGACAC-3′	RP; 5′-CCGGGGTTGTTGTGGTGGTGTC-3′
GAPDH	FP:5′-ACGGGAAACCCATCACCAT-3′	RP; 5′-CCAGCATCACCCCATTTGA-3′

## Data Availability

Data are contained within the article.

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
