# Peer review of "Impairments of Spatial Memory and N-methyl-d-aspartate Receptors and Their Postsynaptic Signaling Molecules in the Hippocampus of Developing Rats Induced by As, Pb, and Mn Mixture Exposure"

_brainsci, 2023, doi:10.3390/brainsci13121715_

Round 1
Reviewer 1 Report
Comments and Suggestions for Authors
BRAINSCI-2739618
Impairments of spatial memory and NMDA receptor and their postsynaptic signaling molecules in the hippocampus of rats induced by developmental As, Pb, Mn – mixture exposure
This study indicates that early-life exposure to As, Pb, Mn, and a mixture of three substances disrupts glutamate signaling and cell signaling pathways in the developing rat hippocampus. Arsenic specifically may induce ERK1/2 activation. This suggests these substances can alter neurochemistry and molecular pathways in the brain when exposure occurs during sensitive developmental periods. However, some methodologies were incomplete, and the logical flow must also be supplemented. Some concerns need to be addressed or clarified before the manuscript can be considered for publication in the journal.
Major comments:
1. What were the specific exposure levels/doses of each substance used in this study? Is there any direct or indirect evidence that can prove the co-exposure of the three substances in real environments?
2. Were both male and female pups examined or only one sex? Please clarify. If only one sex, the tittle should be modified.
3. Was NMDA receptor activation examined as a potential mechanism for the increased glutamate levels? Please clarify.
4. How to ensure the same intake amounts of the substances in test animals since the study used free drinking water administration? Please clarify.
5. Were there any histological or morphological changes observed in the hippocampus in addition to the biochemical changes?
6. The authors should provide the original gel images of western blot in the SI.
Minor comments:
1. The English language needs to check carefully in the revision stage because of there are some careless mistakes in the text. (e.g., the use of abbreviation).
2. The table in the text should have its number and tittle (e.g., Table 1. XXXXX). In addition, please refer to the format of standard statistical tables (three-line tables) to modify the format of this table.
3. Please reorganize your figures (Fig. 2, 3, 4, 6, and 7), as some of them look a bit messy.
4. The authors should modify the reference format according to MDPI’s requirement.
I would like to see the revised manuscript.
Comments on the Quality of English Language
English language fine.
Author Response
Dear Sir, I have incorporate a all suugestions as given by you and also given a resposne to every query. Thanks

Reviewer 2 Report
Comments and Suggestions for Authors
The manuscript discusses the impairments of spatial memory and the NMDA receptor, as well as their postsynaptic signaling molecules in the hippocampus of rats under developmental exposure to a mixture of toxic metals. The study emphasizes the importance of further research on the complex effects of metal mixtures, especially at chronic low levels, which are a significant public health concern. The NMDA receptor and postsynaptic signaling proteins are identified as potential targets for neurotoxic effects caused by exposure to these metals.
The authors presented a high-quality study. I have no comments regarding the experimental design or data analysis. There are only minor comments.
Line 84: Probably a double meaning of "Low levels of As and Pb induce..."
Can be interpreted as neurotoxic effects are caused by the low doses or deficit of metals.
Line 132: The first mentioning of abbreviation GD6-PD60 in the text.
You can guess what is meant, but the abbreviation is not deciphered.
Line 507: "SynGAP is negatively regulated the ERK pathway" or " SynGAP negatively regulates the ERK pathway.." ?
Question: How do authors explain why MM enforces Pb accumulation (Fig 7B) ?
Author Response

(The authors gave the same response as above.)

Reviewer 3 Report
Comments and Suggestions for Authors
The submitted manuscript describes the effects of chronic exposure to low doses of As, Pb and Mn and their mixture on the spatial memory of rats. The authors associate the observed effects with the effect on NMDA receptors. The results are undoubtedly important and interesting, but a number of points need to be clarified.
- The Introduction, in my opinion, is excessively long. It can be reduced by a third. In addition, the authors call arsenic a heavy metal, putting it on a row with cadmium or mercury, although in the classical sense it is not a metal, but a metalloid. Please correct or provide a reference to prove that arsenic is a metal. Additionally, the authors call the concentrations of the studied substances "low", but do not provide ranges defining low or high doses.
- It is better to change the colors of the curves corresponding to the control and the mixture of substances in Figure 1, since upon first look the figure is interpreted in the opposite way.
- The synergistic effect exerted by a mixture of compounds seems quite interesting, but the authors in the discussion consider it only in passing, without putting forward possible hypotheses about the nature of this phenomenon. In addition, an interesting observation is the lack of effect of manganese on spatial memory. What causes this effect?
Comments on the Quality of English Language
Only typos correction is required.
Author Response

(The authors gave the same response as above.)

Round 2
Reviewer 1 Report
Comments and Suggestions for Authors
The authors answered the questions.